# Effect of Mixing Method on Properties of Ethylene Vinyl Acetate Copolymer/Natural Rubber Thermoplastic Vulcanizates

**DOI:** 10.3390/polym12081739

**Published:** 2020-08-04

**Authors:** Nappaphan Kunanusont, Chavakorn Samthong, Fan Bowen, Masayuki Yamaguchi, Anongnat Somwangthanaroj

**Affiliations:** 1Department of Chemical Engineering, Faculty of Engineering, Chulalongkorn University, Bangkok 10330, Thailand; mai2newwood@hotmail.com (N.K.); tao_hp_@hotmail.com (C.S.); 2School of Materials Science, Japan Advanced Institute of Science and Technology, 1-1 Asahidai, Nomi, Ishikawa 923–1292, Japan; paewcu@gmail.com (F.B.); m_yama@jaist.ac.jp (M.Y.)

**Keywords:** ethylene vinyl acetate copolymer, natural rubber, thermoplastic vulcanizate, dynamic vulcanization, mixing method, capillary rheometer

## Abstract

Thermoplastic vulcanizate (TPV) has excellent elastomeric properties and can be reprocessed multiple times. TPV is typically produced by using the dynamic vulcanization (DV) method in which rubber is crosslinked simultaneously with thermoplastics. Peroxide-crosslinked TPV can increase the compatibility between rubber and thermoplastics but loses its reprocessability due to excess crosslinking in the latter. In this work, we overcome this obstacle by using a two-step mixing method to prepare fully crosslinked elastomers of ethylene vinyl acetate copolymer (EVA) and natural rubber (NR). Each sample formulation was prepared with three different mixing methods for comparison: NR-DV, Split-DV, and All-DV. For NR-DV, NR was crosslinked prior to the addition of EVA together with the thermal stabilizer (TS). For Split-DV, a small amount of EVA and NR was crosslinked prior to the addition of EVA and TS. In the All-DV method, EVA and NR were crosslinked, and then TS was added. The appearance and processability of the samples were affected by the degree of crosslinking. NR-DV showed a non-homogeneous texture. Although the samples of the All-DV method appeared homogeneous, their mechanical and rheological properties were inferior to those of the Split-DV method. The mechanical properties of the Split-DV samples were not significantly changed after reprocessing 10 times. Therefore, Split-DV is the preferred method for TPV production.

## 1. Introduction

Recently, environmental issues, especially plastic waste, are a major global concern. There are microparticles of plastics found in soil, animal organs, and the ocean. One of the reasons is caused by disposable plastic, which has been used for packaging [1,2,3,4,5,6,7,8]. Recycling is one way to reduce the plastic waste [2]. Thermoplastic can be recycled because it is molten at high temperature and solidifies again at low temperature. Many products made from rubber cannot be recycled because it is crosslinked and cannot be molten after the process. Therefore, rubber products such as rubber tires and sealants, which are made from crosslinked rubber, cannot be recycled by conventional methods. One way to make use of them is that they need to be ground and mixed with other thermoplastics to be used as fillers [9,10]. However, there is one material that is elastic, similar to rubber, while it can be reprocessed similar to thermoplastics, which is called thermoplastic vulcanizate [11,12].

Thermoplastic vulcanizate (TPV) is a material that has been widely used in automotive industries due to its excellent elastomeric properties (i.e., high toughness) similar to vulcanized rubber as well as its recyclability of thermoplastic behavior. The first commercial TPV was developed by Coran et al., which is a polypropylene (PP)/ethylene–propylene–diene rubber (EPDM) thermoplastic vulcanizate [13,14]. The method that was used to prepare TPV is dynamic vulcanization (DV) process that was invented by Gessler and Haslett [15]. The DV process is to crosslink the elastomer to be fine domains or particles while mixing them with thermoplastic material. In general, the morphology of TPV is that of a sea–island, representing the rubber domain dispersed in the thermoplastic phase. This morphology allows this material to be reprocessed at temperature higher than the melting temperature of the thermoplastic phase. There are several types of crosslinking agents used for the DV process such as sulfur, peroxide, and phenolic resin [16,17,18], which have been used in typical crosslinked elastomers [19,20,21,22,23,24]. Moreover, the curing behavior of the elastomer can be predicted using mathematical techniques such as finite element technique [25], effective close form [26], and forward-backward numerical calculation [27]. For the comparison between sulfur and peroxide, it was found that TPV with peroxide chemicals gave high thermal resistance, while that with sulfur gave high mechanical properties. However, sulfur vulcanization needs many ingredients to achieve a fast and efficient crosslink process. Moreover, sulfur can crosslink with only unsaturated polymer, while peroxide chemicals can crosslink with saturated polymer because it is crosslinked through a radical transfer reaction.

Peroxide is not only used to crosslink with elastomer, it can also increase the compatibility between polymer blends [28,29]. Deetuam et al. [28] improved the properties of poly(lactic acid) (PLA) and natural rubber (NR) blown films by using a small amount of dicumyl peroxide (DCP). They found that adding a small amount of peroxide can increase the compatibility between PLA and NR, resulting in the improvement of the mechanical properties of the PLA/NR system. However, at high DCP content, the NR phase tended to self-crosslink rather than crosslink with PLA, which led to low compatibility between polymers. According to Samthong et al. [29], adding DCP can improve the thermal and mechanical properties of PLA/NBR composites by increasing the compatibility between PLA and acrylonitrile butadiene rubber (NBR), which leads to the small domain size of rubber in the PLA matrix.

For thermoplastic vulcanizates, the external compatibilizers were added because the crosslinking agent selectively crosslinks with only the elastomer phase. Nakaso, et al. [16] studied the effect of compatibilizer types on the properties of NR and HDPE thermoplastic vulcanizates. They found that HDPE modified with phenolic resin gives the optimal mechanical properties and the smallest rubber domain size in the polymer matrix. Intharapat et al. [30] studied the effect of a compatibilizer on the properties of ethylene vinyl acetate copolymer (EVA) and natural rubber (NR) thermoplastic vulcanizate using a sulfur vulcanization system. They found that the smallest domain size of NR was observed in the system with the compatibilizer content of 9 phr. When the more than 9 phr of compatibilizer was added, the compatibilizer agglomerated, resulting in the large domain of NR, which affects the properties of the TPV. Since there is a relationship between the morphology and properties of TPV, the rubber particles should be small to achieve the good mechanical properties of TPV [29,31,32].

Normally, the thermoplastic phase should not be crosslinked, while the elastomer should be fully crosslinked. If the curing is introduced at the same time, it might crosslink both the thermoplastic and elastomer. Therefore, the thermoplastic should be added after the elastomer was crosslinked. Babu et al. [33,34] studied the effect of mixing sequence on the properties of polypropylene (PP)/ethylene octene copolymer (EOC) using peroxide vulcanization. They found that the phase mixing method where the curing agent was pre-mixed with EOC before dynamic vulcanizing with PP showed the optimal mechanical properties of TPV. The radicals from peroxide are not only crosslinked with the rubber, the beta-scission reaction also occurs, which leads to a shorter chain of PP. However, there are a few research concerns about the crosslinking reaction on thermoplastic in TPV, which might affect their properties, especially the flowability of materials.

Herein, the effect of mixing method on the properties of EVA/NR TPVs were studied. The samples were prepared via two-step method, i.e., dynamic vulcanization and further blending. Dynamic vulcanization is to crosslink the NR, which simultaneously mixes with EVA. Then, some of the EVA is further blended to avoid the crosslink reaction of EVA from the dynamic vulcanization step. Three different mixing methods were compared. For NR-DV, NR was crosslinked prior to the addition of EVA together with the thermal stabilizer (TS). For Split-DV, a small amount of EVA and NR was crosslinked prior to the addition of EVA and TS. In the All-DV method, EVA and NR were crosslinked, after which TS was added. The gel content, swelling ratio, and morphology as well as the thermal, dynamic mechanical, rheological, and mechanical properties of EVA/NR TPVs were evaluated to examine the effect of mixing method on the properties of obtained materials. Moreover, the recyclability of TPV was also investigated.

## 2. Materials and Methods 

### 2.1. Materials

Ethylene vinyl acetate copolymer (EVA) (VA content = 18 wt % and MFR = 2.3 g/10min @ 190 °C, 2.16 kg) was purchased from TPI Polene Co., Ltd. (Bangkok, Thailand). Natural rubber sheet (NR) (air-dried sheet type, 59 ML(1+4)100 °C) was purchased from Bothong Rubber Fund Cooperative Ltd. (Chonburi, Thailand). Dicumyl peroxide (DCP) (95 wt % purity) was supplied by Nacalai Tesque, Inc. (Kyoto, Japan). Octadecyl 3-(3,5-di-tert-butyl-4-hydroxyphenyl)propionate (Irganox 1076) and tris(2,4-di-tert-butylphenyl)phosphate (Irgafos 168) as thermal stabilizers were purchased from Ciba Specialty Chemicals Inc. (Basel, Switzerland). Solvents were used as received without further purification. 

### 2.2. Sample Preparation

EVA pellets and NR sheets were dried in a vacuum oven at 60 °C for 4 h to remove moisture. EVA and NR were melt-mixed using an internal mixer with a chamber size of 30 cm^3^ (Labo-Plastomill, Toyoseiki, Nagano, Japan). The mixing method was separated into two steps. Firstly, dynamic vulcanization at 190 °C was performed by pre-melting the polymer for 1 min and then mixing with DCP at 2 phr for 5 min in order to crosslink the desired polymer phase. Afterwards, the dynamically vulcanized polymer was further blended with the remaining polymers and thermal stabilizers (TS) at 130 °C for 6 min to obtain the EVA/NR blends with the final EVA/NR weight ratio of 50/50. Each thermal stabilizer was added at 0.5 phr in order to inhibit the thermal degradation and stop the vulcanization reaction by consuming the free radicals in the systems. Hence, the first step is the vulcanizing mixing step, while the second step is the unvulcanizing mixing step. The rotor speed was fixed at 60 rpm throughout the mixing process. After finishing all mixing steps, the thermoplastic vulcanizates (TPVs) were obtained. Herein, three different dynamic vulcanization protocols were carried out, as shown in the Scheme 1.

#### 2.2.1. Method 1 NR-DV

Only the NR was specifically vulcanized, while the EVA was unvulcanized and was added in the unvulcanizing mixing step.

#### 2.2.2. Method 2 Spilt-DV 

The NR and half of the EVA were vulcanized. Subsequently, the remaining EVA was added in the unvulcanizing mixing step.

#### 2.2.3. Method 3 All-DV 

All the NR and EVA were dynamically vulcanized. Only thermal stabilizers were added in the unvulcanizing mixing step.

The abbreviations of TPVs prepared by NR-DV, Split-DV, and All-DV were denoted as nTPV, sTPV, and aTPV, respectively. Eventually, the obtained blends were compressed into rectangular films using a compression-molding machine at a temperature of 130 °C under 20 bars for 10 min for further characterization. Moreover, to compare the properties between uncured and cured samples, the cured sample was prepared by blending DCP content at 2 phr with polymer using an internal mixer at 100 °C (the decomposition of DCP at this temperature is negligible) with a screw speed of 60 rpm for 6 min. Then, the sample was compressed into a thin film and crosslinked using a compression molding machine at 190 °C under 20 bars for 10 min. The nomenclatures of uncured EVA, uncured NR, cured EVA, and cured NR are uEVA, uNR, cEVA, and cNR, respectively.

### 2.3. Recyclability of TPV

To investigate the recyclability of TPV, the new sample designated as sTPVR0 was melt-mixed in an internal mixer with chamber size of 60 cm^3^ (Charoen Tut, Thailand) at a temperature of 130 °C and rotor speed of 60 rpm for 6 min. Afterward, the obtained sample was compressed into a sheet using a compression-molding machine at temperature of 130 °C under 20 bars for 10 min. The following process is an overall recycle process in this research, as illustrated in Scheme 2. The letter “n” in the Scheme 2 represents the number of recycle times. For example, “sTPVR2” is the sample prepared via the Split-DV method, and it was reprocessed and compressed two times. Especially, there was no addition of thermal stabilizer during this recycle process.

### 2.4. Characterizations

#### 2.4.1. Morphology

Fractured surfaces of EVA/NR sheets were examined by means of a scanning electron microscopy (SEM) (JEOL JSM 5410 LV, Tokyo, Japan) with an acceleration voltage of 15 kV. Samples were coated with osmium tetroxide (OsO_4_) for 30 s to increase the contrast between the EVA phase and NR phase because OsO_4_ selectively stained the double bonds of NR. The NR phase showed a light gray color while the EVA phase showed a dark gray color.

#### 2.4.2. Gel content and Swelling Behavior

To evaluate the degree of crosslinking, gel content, and swelling ratio of the gel, the solvent extraction method was used. During the extraction, solvent (i.e., hot xylene) was diffused into the free volume between the crosslinking points, leading to the swelling behavior of the polymer. The non-crosslinked fraction was extracted from the blends, and the highly crosslinked fraction (gel) remained.

EVA/NR TPVs were packed in the mesh bag and were then extracted in hot xylene for 8 h in order to remove the soluble part of the polymer. Eventually, they were vacuum dried at 80 °C for 6 h to remove the xylene residue. The gel content and swelling ratio of gel were calculated using the following equations:Gel content (%) = (W_3_ − W_1_) / W_0_ × 100(1)
Swelling ratio of gel (%) = (W_2_ − W_1_) / (W_3_ − W_1_) × 100(2)
where

W_0_ = weight of blend sample (g)W_1_ = weight of mesh bag (g)W_2_ = weight of blend sample and mesh bag after extraction (g)W_3_ = weight of blend sample and mesh bag after drying (g)

#### 2.4.3. Differential Scanning Calorimetry (DSC)

Thermal properties (i.e., glass transition temperature (*T*_g_), melting temperature (*T*_m_), crystallization temperature (*T*_c_), enthalpy of crystallization (∆*H*_c_), and enthalpy of melting (∆*H*_m_)) were evaluated by means of differential scanning calorimetry (DSC) (DSC1 module, Mettler Toledo). The sample was firstly heated from −100 to 150 °C at a heating rate of 10 °C/min and maintained at 150 °C for 3 min. Then, it was cooled down to –100°C with a cooling rate of 10 °C/min. Finally, it was secondly heated from −100 to 150 °C again at a heating rate of 10 °C/min. The degree of crystallinity (X_c_) was calculated using the following equation:
X_c_ (%) = [∆*H*_m_ / (ɸ_EVA_ × ∆*H*_m,100%_)] × 100(3)
where
∆*H*_m_ = Enthalpy of melting Φ_EVA_ = mass fraction of EVA in the blend∆*H*_m,100%_ = Enthalpy of melting of polyethylene with 100% crystallinity = 277.1 J/g [35]

#### 2.4.4. Dynamic Mechanical Analysis

Dynamic mechanical properties in the solid state including tensile storage modulus (Eʹ) and tensile loss modulus (E″) were characterized using a dynamic mechanical analyzer (DMA) (Rheogel E4l000, UBM, Kyoto, Japan) from 100 to 120 °C at a heating rate of 2 °C/min and a frequency of 10 Hz in a tension mode. The glass transition temperature (*T*_g_) was obtained from the peak of E″.

#### 2.4.5. Rheological Properties

Steady-state shear viscosity as a function of shear rate at 190 °C was measured using a capillary rheometer (Yasuda Seiki Seisakusho, 140 SAS-2002, Hyogo, Japan) equipped with a circular die of 20 mm in length and 2.0 mm in diameter (L/D = 10). Shear rates were varied in the range of 2–20 s^−1^.

#### 2.4.6. Mechanical Properties

The tensile test was performed following ASTM D412 at room temperature using a tensile testing machine (Little Senstar, Tokyo, Japan) [36]. The dumbbell-shaped samples were punched from the sample sheets. The crosshead speed was 300 mm/min.

For the test of tension set (Figure A1), the dumbbell-shaped samples were stretched to 100% elongation (two times from the initial length) without being broken and were held for 10 min at room temperature. After removing the force, the sample was left for 10 min. The gauge length was measured before and after the test. The tension set was calculated by using the following equation.
Tension set (%) = (L − L_0_) / L_0_ × 100(4)
where
*L* = length after force removal for 10 min (mm)*L*_0_ = initial length (mm)

## 3. Results and Discussion

### 3.1. Appearance and Morphology 

The optical photographs of the compressed film containing 2 phr of DCP prepared by different methods are shown in Figure 1. The sample from the NR-DV method (nTPV) has a rough surface with numerous clearly larger rubber particles than those from the Split-DV method (sTPV) and All-DV method (aTPV), because the rubber phase in the nTPV sample was preferentially vulcanized in the vulcanizing mixing step, leading to the formation of the highly crosslinked NR domains, which were hardly dispersed in the blend. In contrast, the surface of samples prepared by the Split-DV and All-DV methods are relatively smooth, implying good appearance owing to the good dispersion of the NR phase in the matrix. Note that there are small white particles dispersed on the surface of the aTPV sample, which might be the self-crosslinked EVA domain, because the EVA phase was also introduced in the dynamic vulcanization step at the same time as the NR phase. It was confirmed that the mixing method affected the properties of TPV, even though it has the same formulation. Since the nTPV sample contained large rubber particles in the matrix, which is not suitable to use in any application, it was not further examined.

SEM images of the fractured surfaces of sTPV and aTPV with the magnification of 2000 and 5000 times are shown in Figure 2. The samples were stained with osmium tetroxide to increase the contrast between NR and EVA. According to Rajan et al. [37], the peroxide-initiated crosslink reaction for NR only occurs at the secondary carbon atom; thus, the double bonds still remain in the NR chains. Osmium can attach a double bond and improve the contrast between the NR and EVA phases. From the SEM images, the gray area is NR, while the black area is EVA. It was observed that the fractured surface of sTPV was smoother than that of aTPV, as shown in Figure 2a,c. However, the size of the rubber domain of both samples is similar, which can be observed at higher magnification as can be seen in Figure 2b,d. During mixing at the first step, the NR phase was broken by the high shear force of highly viscous EVA into the dispersed domains and distributed throughout the EVA matrix while simultaneously being crosslinked by peroxide radical. However, aTPV, in which all of the EVA is dynamically crosslinked with NR at the first step, might contain more self-crosslinked EVA than that of sTPV. Therefore, the degree of crosslink in the sample should be evaluated to understand the crosslink mechanism of these systems. 

### 3.2. Gel Content and Swelling Ratio

The gel content and swelling ratio of cured EVA, cured NR, and Eva/NR TPVs are shown in Table 1. Gel content (%gel) refers to the amount of highly crosslinked fraction in the TPVs, while the swelling ratio of gel (%swell) refers to the crosslink density. As expected, the uncured sample is totally soluble in hot xylene, and no gel is detected, whereas the cured samples showed a high value of gel content with a low value of swelling ratio of gel. The cured sample was prepared by static vulcanization, the polymer sample is compressed in the mold, and the peroxide radical makes the polymer chain connect together to form the 3D network. For the dynamic vulcanization, the sample was sheared and crosslinked simultaneously. Some of the 3D network can be formed only in the polymer domain. Therefore, the TPVs showed a lower gel content and higher swelling ratio of gel than that of the cured sample. The gel content of sTPV and that of aTPV is similar and it was more than that of nTPV. As can be observed from the sample prepared from the NR-DV method, nTPV has a lower swelling ratio than sTPV and aTPV. The crosslink reaction occurred only in the NR phase for nTPV, while DCP randomly crosslinked both EVA and NR for sTPV and aTPV.

### 3.3. Thermal Properties

The DSC thermograms of uncured EVA, uncured NR, cured EVA, cured NR, and EVA/NR TPVs at the 2^nd^ heating and cooling scans are shown in Figure 3, and their thermal properties are summarized in Table 2. It is found that cured EVA has a lower crystallization temperature (*T*_c_) and melting temperature (*T*_m_) than uncured EVA, because the crosslinked part in EVA restricts the crystallization behavior of EVA in the cooling state, resulting in the formation of imperfect crystals which require less thermal energy to melt. The decrease of melting temperature was also found in peroxide-crosslinked HDPE [38] and crosslinked EVA [39,40,41]. For NR, the crosslinking reaction can be observed from the shift of glass transition temperature (*T*_g_). It is found that *T*_g_ of uncured NR is −65.09 °C, while that of cured NR is −62.53 °C. When the motion of polymer chains is restricted by the crosslink structure, it can be observed by the shift of *T*_g_ to higher temperature [42]. However, the *T*_g_ of both uncured and cured EVA was observed around −30 °C. This result is consistent with the results of Wang and Deng [41]. The crosslink reaction is not affected on *T*_g_ of EVA, which has a VA content lower than 31 wt %.

The EVA/NR TPV obtained from the All-DV method has lower *T*_c_ and *T*_m_ than that from the Split-DV method and NR-DV method, respectively. This is understandable, because in the All-DV method, all EVA was added in the vulcanizing mixing step (first step) in which the peroxide-initiated dynamic vulcanization occurred, leading to a larger amount of self-crosslinked EVA, and in turn, significantly change in the *T*_c_ and *T*_m_ of EVA in the aTPV sample. Conversely, for the NR-DV method, only NR was dynamically vulcanized in the vulcanizing mixing step without EVA, resulting in less significant change in the *T*_c_ and *T*_m_ of EVA in the nTPV sample. Therefore, sTPV, some of whose EVA is added and crosslinked with NR at the first step, showed the *T*_m_ and *T*_c_ values between those of nTPV and aTPV. Moreover, the *T*_g_ of the NR in TPVs was close to the *T*_g_ of the cured NR, which mean that the NR in TPVs was completely crosslinked. However, it was difficult to observe the *T*_g_ of EVA in the TPV system due to the small transition of heat flow compared with a single component system.

### 3.4. Dynamic Mechanical Properties

The storage moduli (E’) and loss moduli (E’’) of uncured and cured EVA and NR as well as EVA/NR TPVs are shown in Figure 4. The E’ and E″ values of uncured NR at the glassy state are higher than those of uncured EVA. In the rubbery region, the E’ of uncured NR was dropped at a temperature of around 110 °C because the sample was broken due to the low thermal stability of the uncured NR. The E’ of uncured EVA greatly decreases when the temperature is higher than its melting temperature (86 °C), while the E’ of the cured samples is constant at a rubbery plateau. The peak of E″ implies the glass transition temperature (*T*_g_) of polymer. The *T*_g_ values of uncured NR and EVA are obviously observed at −58 and −20 °C, respectively. For the cured samples, the *T*_g_ of cured NR shifts to −54 °C because the crosslink network restricts the mobility of the polymer chains. However, the *T*_g_ of cured EVA does not shift to a higher temperature, which is consistent with the DSC results. Moreover, the crosslink of NR and EVA can be observed at the rubbery plateau where both E’ and E″ remained constant. 

For EVA/NR TPVs systems, the obvious peaks of E″ at −50.25 and −51.25 °C were observed for sTPV and aTPV, respectively, which corresponded to the *T*_g_ of NR. However, it is difficult to observe the E″ peak of EVA around −20 °C because the peak height is small compared with the peak of NR. It is worth mentioning that the DMA graph of the sample prepared by the All-DV method (aTPV) has the greatest decreases of both E’ and E″ at a temperature of around 110 °C, because the sample was broken during the DMA test, which might be due to the high self-crosslinking of EVA inducing the phase separation in the blends. Therefore, this reduces the strength of the sample, and the sample broke when receiving the applied force at elevated temperature. The E’ and E″ at the rubbery plateau does not decrease as much as that of uncured EVA at elevated temperature. This shows that the dynamic vulcanization process improves the mechanical properties of the sample at high temperature. When the crosslinking reaction takes place, it restricts the movement of the polymer chains.

### 3.5. Flow Curves and Extruded Strands from Capillary Rheometer

The rheological properties of EVA/NR TPVs are characterized using a capillary rheometer. The flow curves of uncured EVA and TPVs at 190 °C are shown in Figure 5. It shows the non-Newtonian behavior, which is a typical rheological phenomenon of the thermoplastic vulcanizates. The steady-state shear viscosity of aTPV is higher than that of sTPV and uncured EVA. The optical photographs of the strands extruded from the capillary rheometer equipped with a circular die (D = 2 mm) at a shear rate of 120 s^−1^ are shown in Figure 5. The uncured EVA extrudate showed the smooth surface with the highest die swell of 3.0 mm. The sTPV and aTPV extrudate do not exhibit die swell owing to the suitable amount of gel content. However, both the melt fracture and sharkskin behavior were observed in the aTPV. The melt fracture is one type of flow instability that occurs due to the high elasticity of molten polymer, while the sharkskin occurs when the strength of the molten polymer is higher than the stress at the capillary rheometer’s wall [43,44,45]. The gel content of both aTPV and sTPV are similar, as shown in Table 1. However, the aTPV has more self-crosslinked EVA than that of sTPV, as shown by DSC and DMA analysis, which reduces the flowability of the TPVs. Therefore, the TPV should have enough non-crosslinked thermoplastic parts and a suitable crosslinked elastomer to provide the good processability of TPV.

### 3.6. Mechanical Properties

The stress–strain curves of uncured EVA, cured EVA, cured NR, and EVA/NR TPVs at room temperature are shown in Figure 6, and their mechanical properties (i.e., ultimate strength, elongation at break) are summarized in Table 3. The ultimate strength and elongation at break of the samples are obtained from the tensile stress and tensile strain at break, respectively. The uncured and cured EVA showed similar behavior at strain below 50%, as can be seen in Figure 6b. For strain over 50%, the cured EVA showed the higher strength than the uncured one due to its crosslink structure. Unfortunately, uncured NR cannot be tested, because it cannot be prepared as a sheet due to its tackiness. The cured NR showed elastomeric behavior as its nature. It was found that the behavior of sTPV and aTPV was located in between that of EVA and NR. Both sTPV and aTPV also have the same behavior at strain below 100%. However, the behavior of aTPV at strain higher than 100% was similar to that of cured EVA; meanwhile, the sTPV showed behavior similar to the uncured EVA because aTPV contained a greater self-crosslinked structure of EVA than that of sTPV, leading to the phase separation (Figure 1) and poor elongation at break. This result is corresponding to the DMA results in which aTPV was broken more easily than sTPV.

The tension set describes the ability of the material to recover to its original shape after receiving the tension force under the specific temperature and condition. Thus, the tension set directly relates to the elastic properties of the crosslinked polymer. The tension sets of uncured EVA, cured EVA, cured NR, and EVA/NR TPVs at room temperature are summarized in Table 3. It was found that the tension set of cured EVA was higher than that of the uncured EVA, which could be due to the testing method in which the sample was stretched over its elastic region, as can be seen in Figure 6b. Therefore, the cured EVA cannot return to its original shape as the cured NR can. The TPVs show a lower tension set than uncured EVA and cured EVA. It is confirmed that the elastic recovery of TPVs depends only on the crosslinked NR part [16]. Moreover, the tension sets of sTPV and aTPV are in the same range, implying that the mixing method does not affect the tension set value. This result is in agreement with the gel content and swelling ratio.

### 3.7. Recyclability

According to the best mechanical properties among the TPVs in this study, the sTPV was selected to test the recyclability. The sample was reprocessed using the method described in Section 2.3. The tensile test was performed for 0, 5, and 10 cycles. The stress–strain curves of sTPV are shown in Figure 7, and their mechanical properties are summarized in Table 4. It was found that the ultimate strength and elongation at break decreased with an increasing number of recycle times. It might be due to the degradation of polymer during the recycle process [46,47]. However, the stress–strain curves of TPV in the range of 0–200% strain as shown in Figure 7b are similar. From this result, we can infer that the TPV sample can be reprocessed using an internal mixer at 130 °C more than 5 times without the addition of a thermal stabilizer or antioxidant during the recycling process. 

## 4. Conclusions

The suitable method to prepare thermoplastic vulcanizates of ethylene vinyl acetate copolymer and natural rubber (EVA/NR TPVs) by using peroxide vulcanization was successfully identified. Samples were obtained from three mixing methods, i.e., NR-DV, Split-DV, and All-DV, which were compared in terms of (1) appearance and morphology, (2) gel content and swelling ratio, (3) thermal properties, (4) dynamic mechanical properties, (5) rheological properties, and (6) mechanical properties.

Samples obtained from the NR-DV method exhibited a non-homogeneous texture; thus, NR-DV is not a preferred method for TPV production. The gel content and swelling ratio of NR-DV samples (36.73% and 900.85%, respectively) were lower than those of the other two methods, reflecting the lower degree of crosslinking within the samples. The samples of NR-DV had a similar glass transition temperature (*T*_g_) but higher melting temperature (*T*_m_) (86.56 °C) than those of the other two methods. This indicates the ineffectiveness of crosslinked NRs on the crystallization of EVA in the tested samples.

Samples of Split-DV and All-DV had similar appearance and morphology. The gel content and swelling ratio were similar (Split-DV: 47.53%, 1,051.75%; All-DV: 46.45%, 1,135.71%). The *T*_m_ values of the Split-DV samples were higher than those of the All-DV samples (Split-DV: 84.75 °C and All-DV: 82.40 °C), indicating the lower degree of self-crosslinking of EVA. The samples of All-DV were easily broken during the dynamic mechanical tests (at 110 °C), which is attributed to an excess self-crosslinking of EVA. This has some effect on the rheological and mechanical properties of samples obtained from the All-DV method, which resulted in the rough surface of extruded strands and were easily broken when stretched over 400% strain. The Split-DV samples showed overall better quality regarding the dynamic mechanical, rheological, and mechanical properties. Their extruded strands showed smooth surfaces and remained unbroken during the dynamic mechanical test and can be stretched over than 700% strain. As a result, we concluded that Split-DV is the most efficient method for the production of EVA/NR TPVs.

Recyclability was tested on Split-DV samples. The reduction of ultimate strength of Split-DV samples after 10 cycles was twice as much as that after five cycles. However, within the range of below 200% strain, the mechanical properties of recycled products remained almost the same (stress at 200% strain of 0, 5, and 10 cycles are 3.0, 2.8, and 2.6 MPa, respectively). As the normal condition for the use of products, it is not stretched over 200% strain; thus, the recyclability of Split-DV samples is considered acceptable at 10 cycles. It is worth noting that no TS (antioxidant) was added during the recycle process of this study. It is anticipated that mechanical properties could be preserved even more if the thermal stabilizer is added during the recycling process.

Our findings could be applied for other thermoplastics that could be crosslinked with peroxide, for example polyethylene, polystyrene, and polyvinyl alcohol. Furthermore, the morphology development as well as curing profile during the dynamic vulcanization process should be studied. The numerical technique can be an alternative for the studying of this process, by applying a suitable assumption to find the equation constants that can help predict the optimum condition for the curing process.

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
