# Peer review of "Effect of Mixing Method on Properties of Ethylene Vinyl Acetate Copolymer/Natural Rubber Thermoplastic Vulcanizates"

_polymers, 2020, doi:10.3390/polym12081739_

Round 1
Reviewer 1 Report
The main new insight of this work focuses on two-step mixing method, which is used to prepare the fully crosslinked elastomers and to prevent the peroxide crosslinking reaction of thermoplastic part. However, limited new insight is provided with respect to real application, structure and property of this thermoplastic vulcanizate. The author gave a long introduction on TPV, but the reprocessability of the material is not shown. The material should be divided into small fragments and reprocessed under the specific temperature and pressure. In addition, the stress-strain curves before and after reprocessing should be compared. In Page 11, the author said that the tension set describes the ability of material to recover to its original shape. This argument is not convincing. Elongation at break is the ability of material to be stretched and deformed by external force. It can reflect the flexibility of the molecular chain of the material, but it is not directly related to the ability of deformation recovery.
Reviewer 2 Report
Effect of Mixing…
The input of this study is the possibility to recycle the thermoplastic vulcanizate (TPV) in order to reduce the plastic waste. The work consists firstly in the dynamic vulcanization of the NR with DCP at 190°C and after the dynamically vulcanized polymer was further blended at 130°C in order to obtain the EVA/NR blends.
The reviewer has for the authors the following questions:
- What does it mean the dynamic vulcanization of NR at 190°C? The reviewer remembers to the authors the following issues:
- In the vulcanization of NR with DCP the cross-linking efficiency is about 1(see Thomas: J. Appl. Pol. Sci., 6, 613 – 1962: Parks: J. Pol. Sci., 50, 287 – 1961and Scott: J. Pol. Sci. 50, 517 – 1962)
- How is it possible to speak of dynamic vulcanization? What does it mean?
- The study would be increased in value with tests of cure curing following the ASTM D 2084 method and considering and consulting the mathematical methods for NR cross-linking using Sulphur (see and acknowledge with proper discussion the important literature in the field, as for instance Prof. Milani's research group: Rubber Chem. Technol., 88(4), 327 – 2015: J. Math. Chem., 52(2), 464 – 2015 and Polymer Testing, 58, 104 – 2017)
- In the schematic diagram of the mixing method, the questions are:
- Which is the blending technique? did it use brabenber single screw extruder in combination with a static mixer or a static mixer in combination with a double extruder screw.
- In the cases of brabender extruder or double screw extruder wWhich are their profiles?
- It is known that in the mixing method to combine the rubber and plastic materials, the most used is represented in the schematic method with c), but the efficiency of this process is a maximum when the two components have the same melt viscosity. But the melt viscosity of each component depends on molecular parameters as: intrinsic viscosity (dl/g), molecular weight by weight and molecular weight distribution: these parameters have been forgotten in this study.
- In the schematic mixing the total amount of DCP was 3 phr, but in two steps the rubber and plastic are added at the same time but the question is that the DCP is added before in the rubber or in the plastic or where?
- In the conclusions, the authors have found for any product the peculiar characteristics, but they have forgotten to demonstrate the re-cycleability and how many times it is possibleto recycle these new materials.
Reviewer 3 Report
With red ink in the manuscript there are noticed lot of suggestions for correction
The abstract is not clear and should be revised
The Conclusion should be revised, the novelty, main focus and important practical aspect of the research must be added as a first statement.

Round 2
Reviewer 1 Report
This article has been revised as required. Now it can been accepted.
Author Response
Dear reviewer,
Thank you very much.
Kind regards,
Anongnat
Reviewer 2 Report
I am not satisfied by the answers provided by the authors. The huge numerical analyses carried out by others, as required in the original review, should be discussed both in Introduction and Conclusions as possible future research to be carried out. It is hardily acceptable a purely experimental work which seems more a lab report rather than a Journal Paper. I suggest the authors to carefully reply positively to all issues raised and address all properly
Round 3
Reviewer 2 Report
The paper is now acceptable for publication